# PAK4 Is Involved in the Stabilization of PD-L1 and the Resistance to Doxorubicin in Osteosarcoma and Predicts the Survival of Diagnosed Patients

**DOI:** 10.3390/cells13171444

**Published:** 2024-08-28

**Authors:** Junyue Zhang, Yiping Song, Ae-Ri Ahn, Ho Sung Park, See-Hyoung Park, Young Jae Moon, Kyoung Min Kim, Kyu Yun Jang

**Affiliations:** 1Department of Pathology, Jeonbuk National University Medical School, Jeonju 54896, Republic of Korea; yuezai123@naver.com (J.Z.); 18767167431@163.com (Y.S.); arahn@jbnu.ac.kr (A.-R.A.); hspark@jbnu.ac.kr (H.S.P.); kmkim@jbnu.ac.kr (K.M.K.); 2Department of Bio and Chemical Engineering, Hongik University, Sejong 30016, Republic of Korea; 3Department of Biochemistry and Molecular Biology, Jeonbuk National University Medical School, Jeonju 54896, Republic of Korea; yjmoonos@jbnu.ac.kr; 4Department of Orthopedic Surgery, Jeonbuk National University Hospital, Jeonju 54907, Republic of Korea; 5Research Institute of Clinical Medicine, Jeonbuk National University, Jeonju 54896, Republic of Korea; 6Biomedical Research Institute, Jeonbuk National University Hospital, Jeonju 54907, Republic of Korea

**Keywords:** osteosarcoma, PAK4, PD-L1, prognosis, ubiquitination

## Abstract

PAK4 and PD-L1 have been suggested as novel therapeutic targets in human cancers. Moreover, PAK4 has been suggested to be a molecule closely related to the immune evasion of cancers. Therefore, this study evaluated the roles of PAK4 and PD-L1 in the progression of osteosarcomas in 32 osteosarcomas and osteosarcoma cells. In human osteosarcomas, immunohistochemical positivity for the expression of PAK4 (overall survival, *p* = 0.028) and PD-L1 (relapse-free survival, *p* = 0.002) were independent indicators for the survival of patients in a multivariate analysis. In osteosarcoma cells, the overexpression of *PAK4* increased proliferation and invasiveness, while the knockdown of *PAK4* suppressed proliferation and invasiveness. The expression of PAK4 was associated with the expression of the molecules related to cell cycle regulation, invasion, and apoptosis. PAK4 was involved in resistance to apoptosis under a treatment regime with doxorubicin for osteosarcoma. In U2OS cells, PAK4 was involved in the stabilization of PD-L1 from ubiquitin-mediated proteasomal degradation and the in vivo infiltration of immune cells such as regulatory T cells and PD1-, CD4-, and CD8-positive cells in mice tumors. In conclusion, this study suggests that PAK4 is involved in the progression of osteosarcoma by promoting proliferation, invasion, and resistance to doxorubicin and stabilized PD-L1 from proteasomal degradation.

## 1. Introduction

P21-activated kinase (PAK) family members are evolutionarily conserved serine/threonine protein kinases related to various human diseases, such as neurological conditions, cardiac disorders, and cancers [1,2]. In particular, P21-activated kinase 4 (PAK4) is involved in the progression of various human cancers by affecting cellular proliferation, apoptosis, invasiveness, cancer metabolism, and immune regulation [2,3,4]. The expression of PAK4 was increased in various human cancers and related to a poor prognosis for cancer patients, so was proposed as a potential therapy target for cancer patients, including those with sarcomas [4,5,6,7]. In addition, immune checkpoint inhibitors have recently been used extensively, especially those targeting programmed cell death 1 (PD1) and programmed death-ligand 1 (PD-L1) [8]. However, despite some success, some cancer patients did not respond to this therapy or experienced an acceleration of their cancer progression with this therapy [8,9]. Therefore, there are ongoing efforts to enhance the efficacy of immune checkpoint inhibitors and expand the target cancers to which this treatment could be applied [9]. Moreover, PAK4 has been suggested to be closely related to the immune evasion of cancers [3,10], and it has been shown that the suppression of PAK4 enhances the anticancer therapeutic efficacy of immune checkpoint inhibitors [7,11]. Therefore, when considering the role of PAK4 in cancer progression through regulating the cancer microenvironment and the role of PD-L1 in the immune evasion of cancer cells [3,7], there might be a close relationship between PAK4 and PD-L1 in cancer progression and resistance to anticancer therapies. However, despite extensive studies on the roles played by PAK and PD-L1 in human cancers, studies focused on osteosarcoma are limited. Therefore, this study investigated the expression of PAK4 and PD-L1 in human osteosarcoma tissues and further evaluated the role of PAK4 in the progression of osteosarcomas in conjunction with the expression of PD-L1.

## 2. Materials and Methods

### 2.1. Osteosarcoma Specimens and Tissue Samples

Osteosarcomas of the bone that underwent curative resection between January 1998 and December 2012 at Jeonbuk National University Hospital were evaluated in this study. Thirty-two cases with complete medical records, paraffin-embedded tissue blocks, and histologic slides were included in the study. All cases were reviewed according to the 5th edition of the World Health Organization classification of bone tumors [12] and the 8th edition of the American Joint Committee on Cancer staging system [13]. Information on clinicopathologic factors was obtained by reviewing medical records. No patient received preoperative chemotherapy, while 23 patients received adjuvant chemotherapy. This study was conducted with the approval of the Jeonbuk National University Hospital Institutional Review Board (CUH 45 2020-09-045) and was performed according to the Declaration of Helsinki. The approval contained a waiver for written informed consent based on the retrospective and anonymous character of the study.

### 2.2. Immunohistochemical Staining and Scoring in the Tissue Microarray

Immunohistochemical staining for PAK4 and PD-L1 was performed using tissue microarray (TMA) sections from osteosarcomas. In TMA, two 3.0 mm cores were arrayed in each case. The tissue cores were obtained from the original tissue block, and these areas were primarily composed of tumor cells with the highest histologic grade without degenerative change. The TMA sections were deparaffinized and boiled for antigen retrieval in pH 6.0 antigen retrieval buffer (DAKO, 2600 Glostrup, Denmark) in a microwave oven for 20 min. Then, the tissue sections were incubated with anti-PAK4 (1:100, PROTEINTECH, Rosemont, IL, USA) and anti-PD-L1 (1:100, Cell Signaling Technology, Beverly, MA, USA) antibodies overnight at 4 °C. Immunohistochemical staining for PAK4 was evaluated by two pathologists (KYJ and HSP) with consensus and scored without information on the case. The immunohistochemical expressions of PAK4 and PD-L1 in each TMA core were scored by adding the staining intensity (0; no expression, 1; weak expression, 2; intermediate expression, and 3; strong expression) and staining area (0: no stained cells; 1: 1%, 2: 2~10%, 3: 11~33%; 4: 34~66%; and 5: 67~100%) together [4,14,15]. Thereafter, the final score was calculated by adding the scores derived from the two TMA cores together, which ranged from zero to sixteen. 

### 2.3. Cell Lines, Transfection, and Reagents

U2OS and KHOS/NP osteosarcoma cells were used in this study. The U2OS cell line was purchased from the Korean Cell Line Bank (KCLB, Seoul, Republic of Korea), and the KHOS/NP cell line was kindly provided by Chang-Bae Kong (Department of Orthopedic Surgery, Korea Institute of Radiological and Medical Science, Seoul, Republic of Korea). These cells were cultured in Dulbecco’s Modified Eagle Medium (Gibco BRL, Gaithersburg, MD, USA) containing 10% Fetal Bovine Serum (Gibco BRL, Gaithersburg, MD, USA) and 1% penicillin/streptomycin. The vector for *PAK4*-specific shRNA was purchased from Santa Cruz (#SC-39060, Santa Cruz Biotechnology, Santa Cruz, CA, USA). The overexpression of *PAK4* was induced via the transfection of a wild-type *PAK4* (#RG202302, GFP-tagged, OriGene Technologies, Rockville, MD, USA). Lipofectamine 3000 (Invitrogen, Carlsbad, CA, USA) was used for cell transfection. The cells transfected with vectors were used in experiments 24 h after transfection. KPT-9274 (1 µM for 48 h, HY-12793, MedChemExpress, Monmouth Junction, NJ, USA) was used as a PAK4 inhibitor. 

### 2.4. Cell Proliferation Assay

The 3-(4, 5-dimethylthiazol-2-yl)-2, 5-diphenyltetrazolium bromide (MTT) cell proliferation assay (Sigma, Saint Louis, MO, USA) and a colony-forming assay were used to evaluate cellular proliferation. For the MTT assay, 2000 U2OS and 2000 KHOS/NP cells were seeded in 96-well cell culture plates, and optical density was measured at a wavelength of 560 nm. For the colony-forming assay, 2000 U2OS and 1000 KHOS/NP cells were seeded in 6-well plates and grown for seven days. The colonies in the culture plates were stained with 0.01% crystal violet. Their colony-forming ability was evaluated by measuring the area of colonies with the Image J software (version 1.54j, https://imagej.nih.gov/ij (accessed on 14 June 2024)).

### 2.5. In Vitro Trans-Chamber Migration and Invasion Assays

For the migration assay, 5 × 10^4^ U2OS and 2 × 10^4^ KHOS/NP cells in serum-free DMEM were seeded in the upper chamber of a 24-transwell migration chamber (Corning Life Sciences, Tewksbury, MA, USA). For the invasion assay, 2 × 10^5^ U2OS and 1 × 10^5^ KHOS/NP cells in serum-free DMEM were seeded in the upper chamber of a bioCoat Matrigel Invasion Chamber (BD Biosciences, San Jose, CA, USA). The bottom chambers were filled with culture media containing 20% fetal bovine serum as a chemoattractant. The chambers were incubated for 48 h at 37 °C and stained with DIFF-Quik staining solutions (Sysmex, Kobe, Japan). The numbers of migrated and invaded cells at the bottom of the chamber were counted under microscopy. Five × 100 microscopic fields were counted in each chamber. 

### 2.6. Western Blotting, Ubiquitination Analysis, and Immunoprecipitation

The total protein of the cultured osteosarcoma cells was obtained by lysing cells with PRO-PREP Protein Extraction Solution (iNtRON Biotechnology, Seongnam, Republic of Korea). The following primary antibodies were used in this study: PAK4 (#sc-390507, Santa Cruz Biotechnology, Santa Cruz, CA, USA), PD-L1 (#13684, Cell Signaling Technology, Beverly, MA, USA), cleaved PARP1 (#5625, Cell Signaling Technology, Beverly, MA, USA), cleaved caspase 3 (#9661, Cell Signaling Technology, Beverly, MA, USA), cyclin D1 (#2922, Cell Signaling Technology, Beverly, MA, USA), P27 (#sc-528, Santa Cruz Biotechnology, Santa Cruz, CA, USA), BCL2 (#sc-509, Santa Cruz Biotechnology, Santa Cruz, CA, USA), BAX (#sc-20076, Santa Cruz Biotechnology, Santa Cruz, CA, USA), snail (#sc-271977, Santa Cruz Biotechnology, Santa Cruz, CA, USA/#ab180714, Abcam, Cambridge, UK), transforming growth factor β1 (TGF-β1) (#3709, Cell Signaling Technology, Beverly, MA, USA), MMP2 (#sc-13595, Santa Cruz Biotechnology, Santa Cruz, CA, USA), MMP9 (#sc-21733, Santa Cruz Biotechnology, Santa Cruz, CA, USA), FOXO3 (#2880, Cell Signaling Technology, Beverly, MA, USA), phosphorylated FOXO3 (#9464, Cell Signaling Technology, Beverly, MA, USA), and glyceraldehyde 3-phosphate dehydrogenase (GAPDH) (#2118, Cell Signaling Technology, Beverly, MA, USA). For immunoprecipitation, the KHOS/NP and U2OS cells were transfected with an empty vector or shRNA for *PAK4*. In total, 400 µg protein lysates were immunoprecipitated with anti-PAK4 or anti-PD-L1 at 4 °C overnight. The immunocomplexes were pulled down using protein A/G Agarose beads (#20421, Thermo Fisher Scientific, Waltham, MA, USA) and the eluted proteins were immunoblotted with anti-PAK4 (Santa Cruz Biotechnology, Santa Cruz, CA, USA), anti-PD-L1 (Cell Signaling Technology, Beverly, MA, USA), and GAPDH antibodies (Cell Signaling Technology, Beverly, MA, USA). For a ubiquitin proteasomal degradation and ubiquitination analysis, U2OS cells were transfected with an empty vector or shRNA for PAK4 and treated with 30 µmol/L of cycloheximide (CHX, Sigma, Saint Louis, MO, USA) or 30 µmol/L of MG132 (Sigma, Saint Louis, MO, USA) for 30 min to 2 h. The protein lysates were immunoblotted with PD-L1 (Cell Signaling Technology, Beverly, MA, USA) and GAPDH (Cell Signaling Technology, Beverly, MA, USA). In addition, to evaluate the ubiquitination of PD-L1 with the knockdown of *PAK4*, U2OS cells were transfected with an empty vector or shRNA for *PAK4*, and the cells were treated with 30 µmol/L of MG132 for two hours. The total cell lysates were immunoprecipitated with anti-PD-L1 (Cell Signaling Technology, Beverly, MA, USA) and the eluted proteins were immunoblotted with anti-ubiquitin antibodies (#sc-166553, Santa Cruz Biotechnology, Santa Cruz, CA, USA). The membranes were developed by using an enhanced chemiluminescence detection system (Amersham Biosciences, Buckinghamshire, UK), and the images were acquired with a luminescent image analyzer (LAS-4000, Fuji Film, Tokyo, Japan). 

### 2.7. Immunofluorescence Staining

In U2OS and KHOS/NP cells, the cells were fixed with 4% paraformaldehyde. In the formalin-fixed and paraffin-embedded tissue sections, the tissue sections were deparaffinized and antigen retrieval was performed. The cells and tissue sections were incubated with antibodies for PAK4 (#sc-390507, 1:100, Santa Cruz Biotechnology, Santa Cruz, CA, USA), PD-L1 (#13684, 1:200, Cell Signaling Technology, Beverly, MA, USA), PD-L1 (#29122, 1:200, Cell Signaling Technology, Beverly, MA, USA), PD1 (#84651, 1:200, Cell Signaling Technology, Beverly, MA, USA), FoxP3 (#126402, 1:200, BioLegend, San Diego, CA, USA), CD4 (#sc20079, 1:100, Santa Cruz Biotechnology, Santa Cruz, CA, USA), and CD8 (#sc-18860, 1:100, Santa Cruz Biotechnology, Santa Cruz, CA, USA) overnight at 4 °C. Thereafter, the following secondary antibodies were used: Alexa Fluor 488 anti-mouse IgG (#A-10680, Invitrogen, Carlsbad, CA, USA), Alexa Fluor 594 anti-rabbit IgG (#A-11012, Invitrogen, Carlsbad, CA, USA), Alexa Fluor 488 anti-mouse IgG1 (#A-21121, Invitrogen, Carlsbad, CA, USA), Alexa Fluor 594 anti-mouse IgG2b (#A-21145, Invitrogen, Carlsbad, CA, USA), Alexa Fluor 488 anti-rat IgG2b (#53-4031-80, Invitrogen, Carlsbad, CA, USA), Alexa Fluor 594 anti-mouse IgG (#A-11005, Invitrogen, Carlsbad, CA, USA), and Alexa Fluor 647 anti-rabbit IgG (#A-21244, Invitrogen, Carlsbad, CA, USA). The slides were counterstained with DAPI. Images of the immunofluorescence staining were taken with an APEXVIEW™ APX100 All-in-One Microscope (APX100, Olympus, Tokyo, Japan). The number of positively stained cells was counted in three high-power fields in each case, and the sum of the numbers was used for evaluation. The area of one high-power field image was 0.0144 mm^2^. Therefore, 0.0432 mm^2^ was evaluated in each case. 

### 2.8. Quantitative Real-Time PCR with Reverse-Transcription Analysis

The total RNA was extracted with TRIzol reagent (Invitrogen, Carlsbad, CA, USA) according to the manufacturer’s instructions. Reverse transcription to generate cDNA was performed using random hexamer primers from first-strand cDNA synthesis kits (Applied Biosystems, Foster City, CA, USA). A quantitative polymerase chain reaction was performed on 384-well plates using an ABI Prism 7900HT Sequence Detection System (Applied Biosystems, Foster City, CA, USA). The values were normalized to the expression of the GAPDH reference gene. All experiments were performed in triplicate. The primer sequences used in the reverse-transcription polymerase chain reaction are listed in Table 1. 

### 2.9. Tumorigenic Assay

The use of mice was approved by the Institutional Animal Care and Use Committee of Jeonbuk National University (CBNU-2020-0108). To evaluate the in vivo growth of KHOS/NP osteosarcoma cells, eight-week-old male BALB/c nude mice (Orient Bio, Seongnam, Republic of Korea) were used as the orthotopic tumorigenic model. In vivo, tumors were induced by implanting 1 × 10^6^ KHOS/NP cells that were transfected with empty vectors, shRNA for *PAK4*, or plasmid for wild-type *PAK4* into the marrow space of the right proximal tibia under anesthesia. Four mice were used in each group. The tumor volume was measured every seven days with the length × width × height × 0.52 equation. The animals were euthanized at six weeks after cancer cell injection according to humane endpoints using carbon dioxide following sodium pentobarbital anesthesia.

To evaluate the effect of PAK4 on tumor growth in conjunction with the host immunity of the mice, we evaluated the in vivo growth of KHOS/NP cells in male C57BL/6J mice (Orient Bio, Seongnam, Republic of Korea) that maintained host immunity. For this purpose, KHOS/NP tumor masses were generated in four-week-old male BALB/c nude mice (Orient Bio, Seongnam, Republic of Korea). A tumor in the right subcutaneous was induced by injecting 2.5 × 10^6^ KHOS/NP cells transfected with empty vectors, vectors for wild-type *PAK4*, or vectors for shRNA for *PAK4*. Four mice were used in each group and subcutaneous tumor growth was evaluated by measuring the tumor volume every seven days. To evaluate the effect of tumor–host immune interactions on tumor growth, the resected tumors were cut into 2 mm × 2 mm × 2 mm or 0.5 mm × 0.5 mm × 0.5 mm tumor cubes and implanted subcutaneously into four-week-old male C57BL/6J mice (Orient Bio, Seongnam, Republic of Korea). In twelve C57BL/6J mice, four mice in each group were implanted with the 2 mm × 2 mm × 2 mm tumor cubes and maintained until fifteen days, while tumor volume was measured every five days. In another twelve mice, four mice in each group were implanted with the 0.5 mm × 0.5 mm × 0.5 mm tumor cubes; the C57BL/6J mice were euthanized four days after tumor cube implantation. Thereafter, tissue sections were obtained to evaluate the host immune infiltration at the edge of the implanted tumor. The resected tissues were fixed in neutral formalin, embedded in paraffin, and used for hematoxylin and eosin staining and immunofluorescence staining. 

### 2.10. Statistical Analysis

A receiver operating characteristic curve analysis was used to determine immunohistochemical positivity for PAK4 and PD-L1 staining [4,15,16]. The cut-off points were determined at the point with the highest area under the curve to predict the death of patients from osteosarcoma. The prognoses for the overall survival (OS) and relapse-free survival (RFS) of patients were evaluated. The follow-up endpoint was December 2014. An event in the OS analysis was the death of a patient from osteosarcoma, and the duration was calculated from the date of surgery to the date of death or the date of the last follow-up. The patients alive at last contact and those who died from other causes were treated as censored in the OS analysis. An event in the RFS analysis was any type of recurrence or death from osteosarcoma, and the duration was calculated from the date of surgery to the date of the event or the date of the last follow-up. In the RFS analysis, the patients who were alive without relapse at last contact and those died from other causes were treated as censored. The survival analysis was performed through univariate and multivariate Cox proportional hazards regression analyses and a Kaplan–Meier survival analysis. We conducted a multivariate survival analysis using factors that were previously known as potential prognostic indicators of osteosarcoma. The association between the evaluation factors was analyzed with Pearson’s chi-square test, and the comparison of the means between groups was analyzed with the Student’s *t*-test. All experiments were performed in triplicate, and representative data are presented. SPSS software (version 25.0, IBM, Armonk, NY, USA) was used in the statistical analysis, and *p*-values less than 0.05 were considered to be statistically significant. 

## 3. Results

### 3.1. The Expression of PAK4 and PD-L1 Are Associated with Shorter Survival of Osteosarcoma Patients

In human osteosarcomas, PAK4 and PD-L1 were expressed in the cytoplasm and nuclei of tumor cells (Figure 1a). PAK4 positivity and PD-L1 positivity were determined with a receiver operating characteristic curve analysis (Figure 1b). The cut-off point for the immunohistochemical staining scores of both PAK4 expression and PD-L1 expression was twelve (Figure 1b). There was no clinicopathological factor significantly associated with PAK4 and PD-L1 expression (Table 2). However, there was a significant association between PAK4 positivity and PD-L1 positivity (*p* = 0.007) (Table 2).

In the univariate survival analysis, age at diagnosis, T category of stage, PAK4 expression (OS; *p* = 0.007, RFS; *p* = 0.006), and PD-L1 expression (OS; *p* = 0.003, RFS; *p* = 0.003) were significantly associated with OS or RFS (Table 3). The Kaplan–Meier survival curves for PAK4 and PD-L1 expression are presented in Figure 1c. A multivariate survival analysis was performed with the potential prognostic indicators of osteosarcomas: age, TNM stage, T category, N category, M category, and the expression of PAK4 and PD-L1. The multivariate analysis revealed age (RFS; *p* = 0.020), M category of stage (OS; *p* = 0.011), PAK4 positivity (OS; *p* = 0.028), and PD-L1 positivity (RFS; *p* = 0.002) as independent indicators of a poor prognosis for the OS or RFS of osteosarcoma patients. PAK4 positivity indicated a 6.888-fold (95% confidence interval; 1.237–38.367) greater risk of death in the osteosarcoma patients (Table 4). PD-L1 positivity indicated a 5.512-fold (95% confidence interval; 1.863–16.309) greater risk of relapse or death in the osteosarcoma patients (Table 4). In 23 osteosarcoma patients who received adjuvant chemotherapy, the expression of PAK4 (OS; *p* = 0.033, RFS; *p* = 0.016) and PD-L1 (OS; *p* = 0.029, RFS; *p* = 0.019) was significantly associated with both overall survival and relapse-free survival (Figure 1d).

### 3.2. PAK4 Expression Is Associated with the Activity of Proliferation and Invasiveness of Osteosarcoma Cells

In the U2OS and KHOS/NP osteosarcoma cells, the knockdown of *PAK4* significantly inhibited the proliferation of cells (Figure 2a,b). In contrast, the overexpression of *PAK4* significantly increased the proliferation of cells (Figure 2a,b). In addition, the knockdown of *PAK4* significantly decreased the migration and invasion activity of osteosarcoma cells, while the overexpression of *PAK4* significantly increased these activities (Figure 2c,d). Furthermore, the PAK4-mediated activation of the proliferation and invasiveness of osteosarcoma cells was associated with the expression of genes related to cell cycle regulation, apoptosis, and invasiveness. The knockdown of *PAK4* increased the expression of P27 and BAX and decreased the expression of phosphorylated FOXO3, cyclin D1, BCL2, snail, TGF-β1, MMP2, and MMP9 (Figure 2e,f). In contrast, the overexpression of *PAK4* decreased the expression of P27 and BAX and increased the expression of phosphorylated FOXO3, cyclin D1, BCL2, snail, TGF-β1, MMP2, and MMP9 (Figure 2e,f). In an orthotopic tumorigenic model using nude mice, the overexpression of *PAK4* significantly increased the in vivo tumor growth and pulmonary metastasis of KHOS/NP cells (Figure 2g,h). In contrast, in vivo tumor growth and pulmonary metastasis were significantly decreased with the knockdown of PAK4 (Figure 2g,h). There was no metastasis in the liver or kidney any group.

### 3.3. PAK4 Is Involved in Resistance to Doxorubicin of Osteosarcoma Cells

Due to the association of PAK4 positivity with the survival of osteosarcoma patients receiving adjuvant chemotherapy, we investigated the effects of PAK4 on osteosarcoma under treatment with doxorubicin. The knockdown of *PAK4* potentiated the anti-proliferative and apoptotic effects of doxorubicin, and *PAK4* overexpression decreased the effect of doxorubicin on cellular proliferation and apoptosis (Figure 3a–d). The expression of BAX, cleaved PARP1, and cleaved caspase 3 was increased, and the expression of BCL2 decreased with the knockdown of *PAK4* under treatment with 0.1 µM doxorubicin (Figure 3c). The overexpression of *PAK4* increased the expression of BCL2 and decreased the expression of BAX, cleaved PARP1, and cleaved caspase 3 under treatment with doxorubicin (Figure 3c). Apoptotic cells were increased with the knockdown of *PAK4* and decreased with the overexpression of *PAK4* under treatment with 0.1 µM doxorubicin in a flow cytometry analysis (Figure 3d).

### 3.4. PAK4 Is Involved in the Stabilization of PD-L1

Recently, it was shown that the expression of PAK4 is associated with the therapeutic efficacy of immune checkpoint inhibitors [7,11]. Therefore, we evaluated the association between PAK4 and PD-L1 in osteosarcoma cells. The knockdown or overexpression of *PAK4* influenced the expression of the PD-L1 protein (Figure 4a). In addition, KPT9274, a PAK4 inhibitor, decreased the expression of the PD-L1 protein (Figure 4b). However, the expression of the mRNA of PD-L1 was not influenced by the knockdown or overexpression of PAK4 (Figure 4c). Immunofluorescence staining for PAK4 and PD-L1 in osteosarcoma cells showed a positive association between the expression of PAK4 and PD-L1 (Figure 4d). In addition, as PAK4 is expressed in the cytoplasm and nuclei of osteosarcoma cells in human osteosarcoma tissue, the expression of PAK4 in osteosarcoma cells was seen in both the cytoplasm and nuclei of cells (Figure 4d), and its localization did not change with doxorubicin treatment. Together, these data suggest that PAK4 might be involved in the posttranslational stabilization of PD-L1. The immunoprecipitation of PAK4 or PD-L1 showed the direct binding of the PAK4 and PD-L1 proteins (Figure 4e). Under treatment with cycloheximide, the degradation of PD-L1 was increased with the knockdown of *PAK4* (Figure 4f). In addition, the poly-ubiquitination of PD-L1 was increased with the knockdown of *PAK4* (Figure 4g). These results suggest that PAK4 is involved in the posttranslational stabilization of PD-L1 from proteasomal degradation.

### 3.5. PAK4 Expression Is Associated with the Infiltration of Immune Cells in Tumor-Bearing Mice

Based on the positive association between the protein expression of PAK4 and PD-L1, we evaluated the effect of PAK4 on the in vivo growth of KHOS/NP cells in conjunction with the maintained host immunity of male C57BL/6J mice. For this purpose, KHOS/NP cells were implanted in nude mice and grown; thereafter, tumor cubes were re-implanted into the back of C57BL/6J mice (Figure 5a). In both the nude mice and C57BL/6J mice, the overexpression of *PAK4* significantly increased tumor growth, and tumor growth was significantly decreased with the knockdown of *PAK4* (Figure 5b,c). However, in the C57BL/6J mice, the growth of the tumor was delayed five days after tumor cube implantation (Figure 5c). Therefore, immune infiltration in the tumor-implanted mice was evaluated four days after implantation. In the implanted tumors, the expression of PAK4 was associated with PD-L1 expression in immunofluorescence staining (Figure 5d). Furthermore, the overexpression of *PAK4* was significantly associated with an increased infiltration of FOXP3-positive regulatory T cells and PD1-positive cells and a decreased infiltration of CD8-positive cytotoxic cells in peritumoral areas (Figure 5e). In contrast, the knockdown of *PAK4* was significantly associated with a decreased infiltration of FOXP3-positive regulatory T cells and PD1-positive cells and an increased infiltration of CD8-positive cytotoxic cells (Figure 5e). These results suggest that PAK4 might be involved in in vivo tumor progression by affecting host immunity through stabilizing the PD-L1 protein.

## 4. Discussion

This study showed that the expression of PAK4 and the expression of PD-L1 are associated with a shorter survival of osteosarcoma patients. Consistently, the prognostic significance of the expression of PAK4 and PD-L1 has been reported in various human cancers. A higher expression of PAK4 is associated with poor prognoses for patients with cancer in the ovaries [17], lungs [18], prostate [19], and liver [4]. A higher expression of PD-L1 also predicts shorter survival for pancreatic cancer [20], gastric cancer [21], and hepatocellular carcinoma patients [22]. In addition, a higher expression of PD-L1 is an independent indicator of a shorter OS and RFS for people with soft-tissue sarcomas [15]. In a meta-analysis based on 3680 bone and soft-tissue sarcomas, a higher expression of PD-L1 was associated with a shorter survival [23]. Similarly, a higher expression of PD-L1 was associated with an advanced tumor stage and the vascular invasion of osteosarcomas [24]. In this study, despite a small number of osteosarcoma cases, the expressions of PAK4 and PD-L1 were significantly associated with a shorter survival of osteosarcoma patients. Therefore, these results suggest that the expressions of PAK4 and PD-L1 might be used as prognostic indicators of osteosarcomas. However, further study is needed to evaluate the clinicopathological significance of the expression of PAK4 and PD-L1 in more osteosarcoma cases. 

Given the prognostic significance of PAK4 expression in osteosarcomas, this study assessed its impact on the proliferation and invasiveness of osteosarcoma cells. In this study, the expression of PAK4 was closely associated with the proliferation and invasiveness of osteosarcoma cells in both in vitro and in vivo. In the orthotopic tumorigenic model, the overexpression of *PAK4* was associated with more pulmonary metastasis. Consistently, a higher expression of PAK4 increased the proliferation and migration activity of MG63 osteosarcoma cells [25], ovarian cancer cells [17], and prostate cancer cells [19]. In this study, the PAK4-related invasiveness of osteosarcoma cells was associated with the expression of the molecules involved in the epithelial-to-mesenchymal transition (EMT). The EMT is an important hallmark phenotype related to the progression and aggressiveness of human cancers [26]. Even in sarcomas, which are malignant tumors derived from mesenchymal cells, the EMT has been suggested as an important factor in sarcoma progression, and this pattern of molecular change serves to stimulate the invasiveness of cancer cells [27,28]. Consistently, the PAK4-mediated induction of the EMT has been seen in prostate cancer cells [19] and renal tubular epithelial cells [29]. Therefore, this study suggests that PAK4 might be involved in the progression of osteosarcomas by stimulating the EMT. 

In our results, PAK4 expression was related to the survival of osteosarcoma patients who received adjuvant chemotherapy. This result suggests that PAK4 might be related to the efficacy of anticancer therapy. In this study, *PAK4* overexpression was associated with resistance to the doxorubicin-mediated apoptosis of osteosarcoma cells, and osteosarcoma cells that induced the knockdown of *PAK4* were sensitized to doxorubicin-mediated apoptosis. Supportively, PAK4 inhibition reversed the cisplatin resistance of non-small cell lung cancer cells [30] and sensitized pancreatic cancer cells to gemcitabine and multiple chemotherapeutic agents [31,32]. Therefore, although there are limited reports on using PAK4 inhibition in the therapeutic approach to osteosarcoma, it is suggested that using an agent targeting PAK4 might be helpful for those in the poor prognostic group of osteosarcoma patients who express high levels of PAK4.

In addition to the involvement of PAK4 in resistance to apoptosis, PAK4 might be involved in the cancer microenvironment related to the immune evasion of osteosarcoma cells [3,7,10,11]. In this study, there was a significant association between PAK4 and PD-L1 positivity in human osteosarcomas. Furthermore, we presented a mechanism for PD-L1 stabilization in osteosarcoma cells. In U2OS osteosarcoma cells, the knockdown or overexpression of PAK4 did not significantly affect the level of PD-L1 mRNA, but the protein level of PD-L1 decreased with the knockdown of *PAK4* and increased with the overexpression of *PAK4*. In addition, PAK4 directly binds to PD-L1 and delays the proteasomal degradation of PD-L1. The knockdown of PAK4 increased the ubiquitination of PD-L1, which subsequently increased its degradation. Therefore, this study suggests that PAK4 is important in the sustained expression of PD-L1 in osteosarcoma cells. As a cancer marker, as we have shown in this study, PD-L1 expression is significantly associated with the survival of cancer patients [15,20,21,22], and the immunohistochemical expression of PD-L1 can be employed to select the patients who will be subjected to anti-PD-L1 therapy [33]. In addition, continuous studies are ongoing to evaluate the usefulness of anti-PD-L1 therapy in human cancers, for which anti-PD-L1 therapy is still not employed. However, because of the population of cancer patients who did not respond to anti-PD1 therapy, more studies are needed to explain how cancer escapes immune surveillance during anticancer therapies and sustains itself [9,34]. Considering this aspect, we presented interesting results showing that the expression of PD-L1 is regulated by the PAK4-mediated posttranslational stabilization of PD-L1. PAK4 stabilized PD-L1 from proteasomal degradation in osteosarcoma cells. Furthermore, in the in vivo model, a higher expression of PAK4 was associated with the increased peritumoral infiltration of immune cells related to the immune evasion of cancer cells, FOXP3-, and PD1-positive cells. In contrast, the peritumoral infiltration of CD8-positive cytotoxic cells was significantly decreased in mice that were implanted with PAK4-overexpressing KHOS/NP osteosarcoma cells. Similarly, a recent meta-analysis showed that PD-L1 expression was significantly associated with an increased infiltration of PD1-positive cells in bone sarcomas and PD1- and FOXP3-positive cells in soft-tissue sarcomas [23]. In addition, the possibility of the therapeutic efficacy of immunotherapy targeting PD-L1 has been reported in a mouse model of metastatic osteosarcoma [35,36]. Therefore, these results suggest that PAK4 might be involved in the cancer microenvironment related to the immune evasion of osteosarcoma cells.

Another interesting point is that PAK4 behaves similarly to indoleamine 2,3-dioxygenase 1 (IDO1) in its interaction with PD-L1. IDO1 is a heme-containing enzyme, and its heme levels are regulated by nitric oxide [37,38], suggesting that it is a potential prognostic indicator in various cancers [39]. In addition, a high IDO1 RNA expression has been correlated with a high PD-L1 RNA expression [39]. A possible positive association between the immunohistochemical expressions of PD-L1 and IDO1 has also been observed in poorly differentiated thyroid carcinomas [40]. Additionally, in this study, as PAK4 expression was linked to immune cell infiltration, IDO1 expression was similarly correlated with PD-L1 expression and the infiltration of FOXP3-positive cells in melanoma [41]. The therapeutic efficacy of regorafenib in melanoma has also been suggested due to its role in attenuating INFγ-induced PD-L1 and IDO1 expression [42]. Furthermore, targeting IDO1 in combination with immune checkpoint inhibitors has been proposed as a novel therapeutic approach for human cancers [43]. Therefore, although we did not evaluate IDO1 expression in osteosarcoma, there may be crosstalk between PAK4 and IDO1, which could represent a novel therapeutic target in human cancers. However, further studies are needed to explore this relationship.

## 5. Conclusions

In conclusion, this study demonstrates that the high expressions of PAK4 and PD-L1 are indicators of a poor prognosis in osteosarcoma patients. Additionally, PAK4 is involved in the progression of osteosarcoma by promoting proliferation, invasion, and resistance to anticancer therapies. In particular, PAK4-mediated resistance to anticancer therapy might be mediated by resistance to apoptosis and the induction of immune evasion by stabilizing PD-L1 expression. Therefore, this study suggests that targeting the PAK4 might be a novel therapeutic approach for osteosarcoma patients with tumors expressing high levels of PAK4 and PD-L1.

## Figures and Tables

**Figure 1 cells-13-01444-f001:**
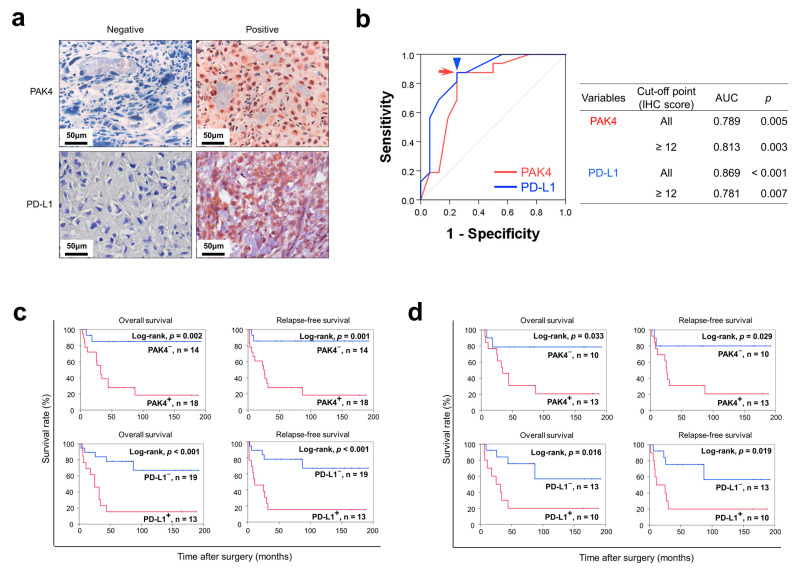
Immunohistochemical expression of PAK4 and PD-L1 in human osteosarcoma tissue and survival analysis. (**a**) Immunohistochemical expression of PAK4 and PD-L1 in human osteosarcoma tissue. Original magnification, ×400. (**b**) Statistical analysis to determine cut-off points. In receiver operating characteristic curve analysis, the cut-off point for both the immunohistochemical staining scores of PAK4 expression (red arrowhead) and PD-L1 expression (blue arrow) was twelve. (**c**) Kaplan–Meier survival analysis according to PAK4 and PD-L1 expression for overall survival and relapse-free survival in 32 osteosarcoma patients. (**d**) Kaplan–Meier survival analysis for overall survival and relapse-free survival according to PAK4 and PD-L1 expression in 23 osteosarcoma patients who received postoperative chemotherapy.

**Figure 2 cells-13-01444-f002:**
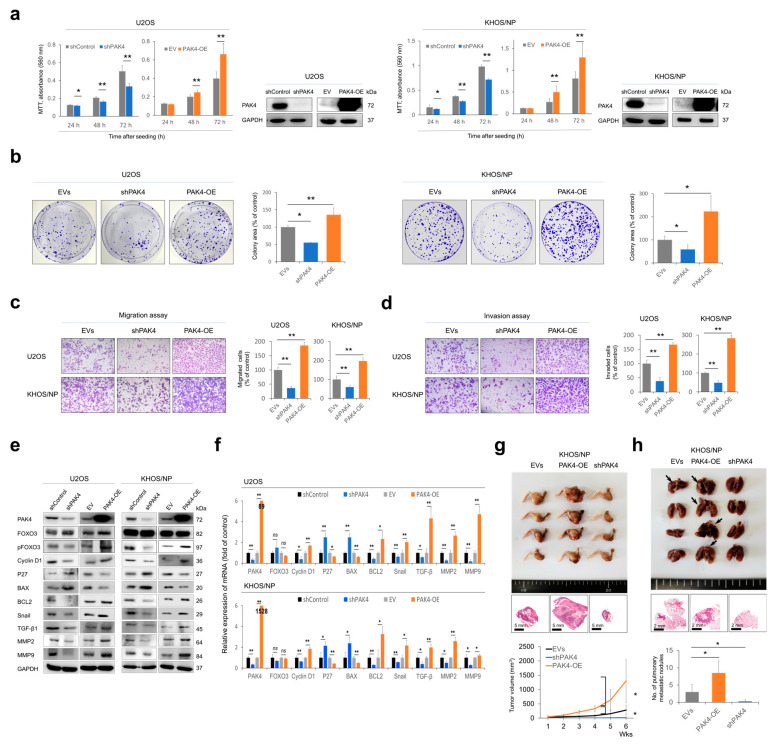
The effects of PAK4 expression on proliferation and invasiveness in osteosarcoma cells. (**a**,**b**) MTT proliferation assays (**a**) and colony-forming assay (**b**) after knockdown or overexpression of *PAK4* in U2OS and KHOS/NP osteosarcoma cells. Colony-forming assays were performed after knockdown or overexpression of *PAK4* in osteosarcoma cells. U2OS (3 × 10^3^) and KHOS/NP (3 × 10^3^) cells were seeded in culture plates for seven days. Western blot for PAK4 and GAPDH was performed to show knockdown and overexpression of *PAK4* in osteosarcoma cells. (**c**,**d**) Migration (**c**) and invasion (**d**) assay after knockdown or overexpression of *PAK4* in U2OS and KHOS/NP osteosarcoma cells. The migration assay was performed by seeding 5 × 10^4^ U2OS and 5 × 10^4^ KHOS/NP cells in the upper chamber for 48 h. The invasion assay was performed by seeding 1 × 10^5^ U2OS and 1 × 10^5^ KHOS/NP cells in the upper chamber for 48 h. (**e**) Western blotting for PAK4, FOXO3, phosphorylated FOXO3 (pFOXO3), cyclin D1, P27, BAX, BCL2, snail, TGF-β1, MMP2, and MMP9 after knockdown or overexpression of PAK4 in U2OS and KHOS/NP osteosarcoma cells 24 h after transfection. (**f**) Quantitative reverse-transcription polymerase chain reaction for PAK4, FOXO3, cyclin D1, P27, BAX, BCL2, snail, TGF-β1, MMP2, and MMP9 after knockdown or overexpression of PAK4 in osteosarcoma cells. (**g**) Gross and histologic findings of resected tumors grown in BALB/c nude mice by implanting 1 × 10^6^ KHOS/NP cells that were transfected with empty vectors, shRNA for *PAK4*, or plasmid for wild-type *PAK4* into the marrow space of the right proximal tibia. The tumor volume was measured every seven days with the length × width × height × 0.52 mm^3^ equation. The mice were euthanized six weeks after tumor implantation. Resected tumors were H&E stained. (**h**) Gross and histologic findings of pulmonary metastatic nodules in BALB/c nude mice. Arrows indicate metastatic nodules. * *p* < 0.05; ** *p* < 0.001; EVs, empty vectors; PAK4-OE, vector for wild-type *PAK4*; shControl, control vector for shRNA; shPAK4, vector for shRNA for *PAK4*.

**Figure 3 cells-13-01444-f003:**
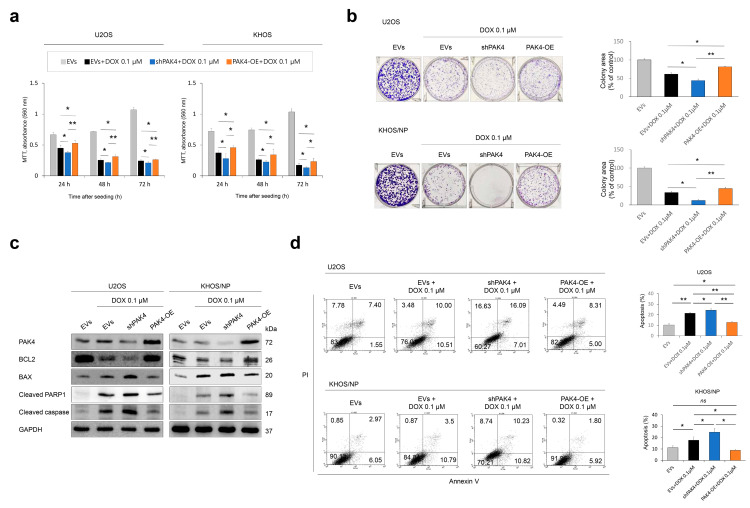
The effects of PAK4 expression on proliferation and apoptosis of osteosarcoma cells under treatment of doxorubicin. (**a**,**b**) MTT proliferation assays (**a**) and colony-forming assay (**b**) after knockdown or overexpression of *PAK4* in U2OS and KHOS/NP osteosarcoma cells treated with 0.1 µM doxorubicin. In colony-forming assay, U2OS (3 × 10^3^) and KHOS/NP (3 × 10^3^) cells were seeded in culture plates for one week. (**c**,**d**) Western blotting (**c**) and Annexin V flowcytometric analysis for apoptosis (**d**) after knockdown or overexpression of *PAK4* in osteosarcoma cells under treatment with 0.1 µM doxorubicin. Doxorubicin was applied to osteosarcoma cells 24 h after transfection. * *p* < 0.05; ** *p* < 0.001; *ns*, not significant; EVs, empty vectors; PAK4-OE, vector for wild-type *PAK4*; shPAK4, vector for shRNA for *PAK4*; DOX, doxorubicin.

**Figure 4 cells-13-01444-f004:**
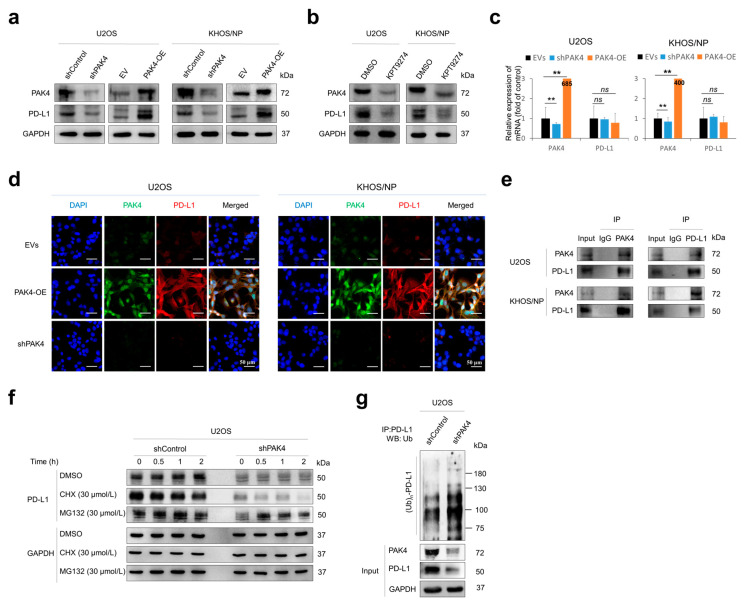
PAK4 is involved in the stabilization of PD-L1 protein in osteosarcoma cells. (**a**) Western blot for PAK4 and PD-L1 after knockdown or overexpression of *PAK4* in osteosarcoma cells. (**b**) Western blot for PAK4 and PD-L1 in osteosarcoma cells after treatment with PAK4 inhibitor KPT9274 (1 µM for 48 h). (**c**) Quantitative reverse-transcription polymerase chain reaction for PAK4 and PD-L1 after knockdown or overexpression of *PAK4* in U2OS and KHOS/NP osteosarcoma cells. (**d**) Immunofluorescence staining of U2OS and KHOS/NP osteosarcoma cells after knockdown or overexpression of *PAK4*. The cells were incubated with primary antibodies for PAK4 and PD-L1. Thereafter, the slides were incubated with Alexa Fluor 488 anti-mouse IgG (green) or Alexa Fluor 594 anti-rabbit IgG (red) and counterstained with 4′,6-diamidino-2-phenylindole (DAPI, blue). Images were taken with a Zeiss LSM 880 with Airyscan confocal microscope. (**e**) U2OS and KHOS/NP osteosarcoma cell lysates were immunoprecipitated with PAK4 or PD-L1 and immunoblotted with PAK4 and PD-L1. (**f**) U2OS cells were transfected with control vector or shRNA for *PAK4* and treated with 30 µmol/L cycloheximide or 30 µmol/L MG132 for 0.5 to 2.0 h. Thereafter, protein lysates were immunoblotted with PD-L1 and GAPDH. (**g**) Total protein lysate from U2OS cells transfected with empty vector or shRNA for *PAK4* and treated with 30 µmol/L MG132 for two hours was immunoprecipitated with PD-L1 and immunoblotted with anti-ubiquitin antibodies. ** *p* < 0.001; *ns*, not significant; CHX, cycloheximide; EV, empty vector; EVs, empty vectors; IP, immunoprecipitation; PAK4-OE, vector for wild-type *PAK4*; shControl, control vector for shRNA; shPAK4, vector for shRNA for *PAK4*; WB, western blot.

**Figure 5 cells-13-01444-f005:**
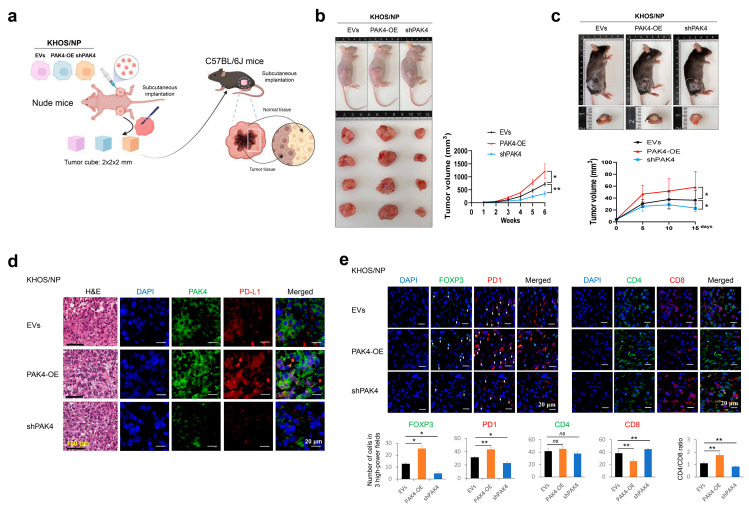
In vivo growth of KHOS/NP osteosarcoma cells in BALB/c nude mice and infiltrations of immune cells in C57BL/6J mice. (**a**) HOS/NP osteosarcoma cells (2.5 × 10^6^) transfected with empty vectors, vector for shRNA for *PAK4*, or vector for wild-type *PAK4* were implanted subcutaneously in the back of BALB/c nude mice and grown for six weeks. Donor tumor blocks sized 2 mm × 2 mm × 2 mm from the resected tumor of BALB/c nude mice were subcutaneously implanted in the backs of C57BL/6J mice. The figure was created with BioRender.com (https://www.biorender.com (accessed on 10 January 2024)). (**b**) Tumor growth in BALB/c nude mice. (**c**) The growth of tumor block in the back of C57BL/6J mice. Tumor volumes were calculated as “length × width × height × 0.52”. (**d**) Histologic findings and immunofluorescence staining for PAK4 (green) and PD-L1 (red) in the resected tumor grown in C57BL/6J mice. (**e**) Immunofluorescence staining and quantification for FOXP3, PD1, CD4, CD8, and CD4/CD8 ratio. Tumor cubes (0.5 mm × 0.5 mm × 0.5 mm) derived from the KHOS/NP tumor cells grown subcutaneously in BALB/c nude mice were implanted subcutaneously in C57BL/6J mice. Four days after the implantation of tumor cubes, the tumors were resected and evaluated with immunofluorescence staining for FOXP3 (green), PD1 (red), CD4 (green), and CD8 (red). The number of positively stained cells was counted in three high-power fields in each case, and the sum of the numbers was used for evaluation. The area of one high-power field image was 0.0144 mm^2^. Therefore, 0.0432 mm^2^ was evaluated in each case. * *p* < 0.05; ** *p* < 0.001; *ns*, not significant; EVs, empty vectors; PAK4-OE, vector for wild-type *PAK4*; shPAK4, vector for shRNA for *PAK4*.

**Table 1 cells-13-01444-t001:** Primer sequences used for a quantitative real-time polymerase chain reaction.

Gene	Primer Sequence		Product Size	Accession Number
*PAK4*	forward	5′-GGACATCAAGAGCGACTCGAT-3′	113	NM_001014831.3
	reverse	5′-CGACCAGCGACTTCCTTCG-3′		
*PD-L1*	forward	5′-GGACAAGCAGTGACCATCAAG-3′	235	NM_001267706.2
	reverse	5′-CCCAGAATTACCAAGTGAGTCCT-3′		
*FOXO3*	forward	5′-CGGACAAACGGCTCACTCT-3′	150	NM_001455
	reverse	5′-GGACCCGCATGAATCGACTAT-3′		
*CCND1* (Cyclin D1)	forward	5′-GAGGAAGAGGAGGAGGAGGA-3′	236	NM_053056.2
	reverse	5′-GAGATGGAAGGGGGAAAGAG-3′		
*P27*	forward	5′-TCTACTGCGTGGCTTGTCAG-3′	240	AB001740.1
	reverse	5′-CTGTATTTGGAGGCACAGCA-3′		
*BAX*	forward	5′-CCCGAGAGGTCTTTTTCCGAG-3′	155	NM_138763
	reverse	5′-CCAGCCCATGATGGTTCTGAT-3′		
*BCL2*	forward	5′-GGTGGGGTCATGTGTGTGG-3′	89	NM_000657
	reverse	5′-CGGTTCAGGTACTCAGTCATCC-3′		
*SNAL1* (Snail)	forward	5′-ACCCCACATCCTTCTCACTG-3′	217	NM_005985.3
	reverse	5′-TACAAAAACCCACGCAGACA-3′		
*TGF-* *β1*	forward	5′-CCCACAACGAAATCTATGACAA-3′	246	NM_000660.7
	reverse	5′-AAGATAACCACTCTGGCGAGTG-3′		
*MMP2*	forward	5′-GATACCCCTTTGACGGTAAGGA-3′	112	NM_004530
	reverse	5′-CCTTCTCCCAAGGTCCATAGC-3′		
*MMP9*	forward	5′-TGTACCGCTATGGTTACACTCG-3′	97	NM_004994
	reverse	5′-GGCAGGGACAGTTGCTTCT-3′		
*GAPDH*	forward	5′-AACAGCGACACCCACTCCTC-3′	258	NM_001256799.1
	reverse	5′-GGAGGGGAGATTCAGTGTGGT-3′		

Web link to accession numbers: https://www.ncbi.nlm.nih.gov/gene (accessed on 10 January 2024).

**Table 2 cells-13-01444-t002:** Association between clinicopathologic variables and the expression of PAK4 and PD-L1 in 32 osteosarcomas.

Characteristics		No.	PAK4		PD-L1	
			Positive	*p*	Positive	*p*
Age, years	<30	24	12 (50%)	0.217	9 (38%)	0.533
	≥30	8	6 (75%)		4 (50%)	
Sex	Male	20	13 (65%)	0.198	8 (40%)	0.926
	Female	12	5 (42%)		5 (42%)	
TNM stage	IIA	15	8 (53%)	0.755	5 (33%)	0.430
	IIB, III, IV	17	10 (59%)		8 (47%)	
T category	T1	16	8 (50%)	0.476	5 (31%)	0.280
	T2, T3	16	10 (63%)		8 (50%)	
N category	N0	30	17 (57%)	0.854	12 (40%)	0.780
	N1	2	1 (50%)		1 (50%)	
M category	M0	27	15 (56%)	0.854	11 (41%)	0.975
	M1	5	3 (60%)		2 (40%)	
PD-L1	Negative	19	7 (37%)	0.007		
	Positive	13	11 (85%)			

TNM, tumor node metastasis.

**Table 3 cells-13-01444-t003:** Univariate survival analysis in 32 osteosarcomas.

Characteristics	No.	OS		RFS	
		HR (95% CI)	*p*	HR (95% CI)	*p*
Age, years, ≥30 (vs. <30)	8/32	3.071 (1.105–8.532)	0.031	3.139 (1.128–8.739)	0.029
Sex, male (vs. female)	20/32	0.874 (0.298–2.567)	0.807	0.768 (0.263–2.246)	0.630
TNM stage, ≥IIB (vs. IIA)	17/32	2.865 (0.983–8.349)	0.054	2.912 (0.997–8.507)	0.051
T category, T2 and T3 (vs. T1)	16/32	3.009 (1.035–8.750)	0.043	3.142 (1.079–9.150)	0.036
N category, N1 (vs. N0)	2/32	3.830 (0.441–33.245)	0.223	2.047 (0.255–16.465)	0.501
M category, M1 (vs. M0)	5/32	3.297 (0.904–12.025)	0.071	2.553 (0.703–9.277)	0.155
PAK4, positive (vs. negative)	18/32	7.646 (1.726–33.875)	0.007	7.981 (1.801–35.369)	0.006
PD–L1, positive (vs. negative)	13/32	5.195 (1.768–15.260)	0.003	5.157 (1.768–15.039)	0.003

OS, overall survival; RFS, relapse-free survival; HR, hazard ratio; 95% CI, 95% confidence interval; TNM, tumor node metastasis.

**Table 4 cells-13-01444-t004:** Multivariate survival analysis in 32 osteosarcomas.

Characteristics	OS		RFS	
	HR (95% CI)	*p*	HR (95% CI)	*p*
Age, years, ≥30 (vs. <30)			3.502 (1.221–10.045)	0.020
M category, M1 (vs. M0)	8.491 (1.622–44.448)	0.011		
PAK4, positive (vs. negative)	6.888 (1.237–38.367)	0.028		
PD-L1, positive (vs. negative)	2.978 (0.902–9.835)	0.073	5.512 (1.863–16.309)	0.002

OS, overall survival; RFS, relapse-free survival; HR, hazard ratio; 95% CI, 95% confidence interval. Multivariate survival analysis was performed with the potential prognostic indicators of osteosarcomas: age, TNM stage, T category, N category, M category, and the expression of PAK4 and PD-L1.

## Data Availability

The datasets used and/or analyzed during the current study are available from the corresponding author upon reasonable request.

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
