# Peer review of "PAK4 Is Involved in the Stabilization of PD-L1 and the Resistance to Doxorubicin in Osteosarcoma and Predicts the Survival of Diagnosed Patients"

_cells, 2024, doi:10.3390/cells13171444_

Round 1

Reviewer 1 Report

Comments and Suggestions for Authors

Dear Authors,

The manuscript I have called to review describes the use of PAK4 as a diagnostic marker and at the same time provides a possible mechanism for its way of action.

I think this manuscript should be accepted as it contains excellent issues to be further explored, albeit we all know osteosarcoma is a very bad and insidious disease.

However, before being accepted I would like the authors to clarify some points:

1.     Please specify in which way PAK4 silencing has been obtained. If it was reached by transient transfection, please specify how many days after transfection the experiments were performed. On the contrary, if they are stable clones, please add the specification in the methods section.

2.     FIGURE 2E: Please provide after how many days from seeding, cells were treated.

3.     FIGURE 3A-B: Please provide the time of treatment with doxorubicin, and how many days after transfection these experiments were performed.

4.     FIGURE 4C: Please indicate what captions stand for.

5.     Finally, where is PAK4 supposed to localize within the cells? Is this localization confirmed in cancer cells? Does PAK4 change this localization following doxorubicin treatment? Please provide evidence for this, by immunofluorescence analysis on treated cells.

Author Response

Response to reviewer 1

We thank the reviewer for these insightful comments.

Comments from Reviewer #1:

Dear Authors,

The manuscript I have called to review describes the use of PAK4 as a diagnostic marker and at the same time provides a possible mechanism for its way of action.

I think this manuscript should be accepted as it contains excellent issues to be further explored, albeit we all know osteosarcoma is a very bad and insidious disease.

However, before being accepted I would like the authors to clarify some points:

We thank the reviewer for these insightful comments.

  1. Please specify in which way PAK4 silencing has been obtained. If it was reached by transient transfection, please specify how many days after transfection the experiments were performed. On the contrary, if they are stable clones, please add the specification in the methods section.

We thank the reviewer for this comment. In this study, we have used transient transfection to induce silencing or overexpression of PAK4. In every experiment, we have transfected and used cells 24 hours after transfection.

In response to the comment of the reviewer, we have revised the manuscript as follows:

“The cells transfected with vectors were used in experiments 24 hours after transfection.”

  1. FIGURE 2E: Please provide after how many days from seeding, cells were treated.

We thank the reviewer for this comment. As we responded to the previous comment of the reviewer, we evaluated it 24 hours after transfection.

In response to the comment of the reviewer, we have revised the manuscript as follows:

“… in U2OS and KHOS/NP osteosarcoma cells 24 hours after transfection.”

  1. FIGURE 3A-B: Please provide the time of treatment with doxorubicin, and how many days after transfection these experiments were performed.

We thank the reviewer for this comment. As we responded to the previous comment of the reviewer, we evaluated it 24 hours after transfection. In this experiment, 24 hours after transfection, the cells were treated with doxorubicin.

In response to the comment of the reviewer, we have revised the manuscript as follows:

“Doxorubicin was treated to osteosarcoma cells 24 hours after transfection.”

  1. FIGURE 4C: Please indicate what captions stand for.

 We thank the reviewer for this comment. There is a mistake in including Figure 4 in the manuscript. We have fixed this mistake and revised it to show Figure 4c clearly in the revised manuscript.

  1. Finally, where is PAK4 supposed to localize within the cells? Is this localization confirmed in cancer cells? Does PAK4 change this localization following doxorubicin treatment? Please provide evidence for this, by immunofluorescence analysis on treated cells.

We thank the reviewer for this comment. As we have shown in Figure 4d EVs, PAK4 is expressed in nuclei and cytoplasm of the U2OS and KHOS/NP cells. In addition, in response to the comments of the reviewer, we rechecked the expression of PAK4 in osteosarcoma cells with or without treatment of doxorubicin. In both cells without doxorubicin treatment and doxorubicin treatment, PAK4 was expressed in the cytoplasm and nuclei of cells (below figure). The subcellular localization of PAK4 was not changed with doxorubicin treatment.

Concerning this point, we have revised the manuscript as follows:

“In addition, as PAK4 expressed in the cytoplasm and nuclei of osteosarcoma cells in human osteosarcoma tissue, the expression of PAK4 in osteosarcoma cells was seen in both the cytoplasm and nuclei of cells (Figure 4d) and its localization was not changed with doxorubicin treatment (data not shown).”

Reviewer 2 Report

Comments and Suggestions for Authors

Very interesting findings by the authors. PAK4 behaves very similar to IDO1 in terms of their crosstalk with PD-L1. Could there be any crosstalk between PAK4 and IDO1? The authors must discuss the roles of IDO1 and PD-L1 too in their discussion section as to their similarities with PAK4 in cancer (10.1126/scitranslmed.3006504, 10.1007/s12022-018-9514-y, 10.1158/1078-0432.CCR-18-2840, 10.1016/j.ctrv.2022.102461, 10.1016/j.isci.2024.109632 etc). IDO1 is a very important prognostic marker in various cancers. It is a heme containing enzyme, its heme insertion mechanism was recently discovered (10.1016/j.freeradbiomed.2022.01.008) and that its heme levels are regulated by nitric oxide (10.1016/j.jbc.2023.104753). The authors should discuss these papers in their discussion section which would enrich the manuscript further and draw a parallel between PAK4 and IDO1 with respect to their interaction with PD-L1. Thank you.

Author Response

Response to reviewer 2

We thank the reviewer for these insightful comments.

Comments from Reviewer #2:

Very interesting findings by the authors. PAK4 behaves very similar to IDO1 in terms of their crosstalk with PD-L1. Could there be any crosstalk between PAK4 and IDO1? The authors must discuss the roles of IDO1 and PD-L1 too in their discussion section as to their similarities with PAK4 in cancer (10.1126/scitranslmed.3006504, 10.1007/s12022-018-9514-y, 10.1158/1078-0432.CCR-18-2840, 10.1016/j.ctrv.2022.102461, 10.1016/j.isci.2024.109632 etc). IDO1 is a very important prognostic marker in various cancers. It is a heme containing enzyme, its heme insertion mechanism was recently discovered (10.1016/j.freeradbiomed.2022.01.008) and that its heme levels are regulated by nitric oxide (10.1016/j.jbc.2023.104753). The authors should discuss these papers in their discussion section which would enrich the manuscript further and draw a parallel between PAK4 and IDO1 with respect to their interaction with PD-L1. Thank you.

We thank the reviewer for this comment. In response to the comment of the reviewer, concerning this point, we have revised the manuscript as follows:

“Another interesting point is that PAK4 behaves similarly to indoleamine 2,3-dioxygenase 1 (IDO1) in its interaction with PD-L1. IDO1 is a heme-containing enzyme, and its heme levels are regulated by nitric oxide [37,38], suggesting it is a potential prognostic indicator in various cancers [39]. In addition, high IDO1 RNA expression has been correlated with high PD-L1 RNA expression [39]. A possible positive association between the immunohistochemical expressions of PD-L1 and IDO1 has al-so been observed in poorly differentiated thyroid carcinomas [40]. Additionally, in this study, as PAK4 expression was linked to immune cell infiltration, IDO1 expression was similarly correlated with PD-L1 expression and the infiltration of FOXP3-positive cells in melanoma [41]. The therapeutic efficacy of regorafenib in melanoma has also been suggested due to its role in attenuating INFγ-induced PD-L1 and IDO1 expression [42]. Furthermore, targeting IDO1 in combination with immune checkpoint inhibitors was proposed as a novel therapeutic approach for human cancers [43]. Therefore, although we did not evaluate IDO1 expression in osteosarcoma, there may be crosstalk between PAK4 and IDO1, which could represent a novel therapeutic target in human cancers. However, further studies are needed to explore this relationship.”